# Assessing the validity of saliency maps for abnormality localization in medical imaging

**Nishanth Thumbavanam Arun**[*1]

**Nathan Gaw**[*2]

**Praveer Singh**[*1]

**Ken Chang**[*1]

**Katharina Viktoria Hoebel**[1]

**Jay Patel**[1]

**Mishka Gidwani**[1]

**Jayashree Kalpathy-Cramer**[1]                    KALPATHY@NMR.MGH.HARVARD.EDU

[1] *Athinoula A. Martinos Center for Biomedical Imaging, Department of Radiology, Massachusetts General Hospital, Boston, MA, USA*

[2] *Arizona State University-Mayo Clinic Center for Innovative Imaging, School of Computing, Informatics, and Decision Systems Engineering, Tempe, AZ, USA*

## Abstract

Saliency maps have become a widely used method to assess which areas of the input image are most pertinent to the prediction of a trained neural network. However, in the context of medical imaging, there is no study to our knowledge that has examined the efficacy of these techniques and quantified them using overlap with ground truth bounding boxes. In this work, we explored the credibility of the various existing saliency map methods on the RSNA Pneumonia dataset. We found that GradCAM was the most sensitive to model parameter and label randomization, and was highly agnostic to model architecture.

**Keywords:** Saliency maps, localization, deep learning.

## 1. Introduction

Saliency maps have become a popular approach for post-hoc interpretability of Convolutional Neural Networks (CNNs). (Adebayo et al., 2018) These maps are designed to highlight the salient components of the input images that are important to the model prediction. As a result, many deep learning medical imaging studies have used saliency maps to rationalize model prediction and provide localization. (Rajpurkar et al., 2017; Bien et al., 2018; Mitani et al., 2019) However, the validity of saliency maps has been called into question in a recent study showing that many popular saliency map approaches are not sensitive to model weight or label randomization for models evaluated on several datasets. (Adebayo et al., 2018) In this study, we extend this work by evaluating popular saliency map methods

---

[*] Contributed equally

both quantitatively and qualitatively for classification models trained on the RSNA Pneumonia dataset. (Shih et al., 2019) Specifically, we assess the performance of these methods in localizing abnormalities in medical imaging by quantifying overlap with ground truth bounding boxes. Furthermore, we assess the effect of model weight and label randomization on localization performance. Lastly, we empirically study repeatability of the saliency maps, both within the same model architecture and across different model architectures.

## 2. Methods and Results

### 2.1. Model and Data Randomization

The saliency methods examined in our experiments are Gradient Explanation (Simonyan et al., 2013), Smoothgrad Integrated Gradients (IG) (Sundararajan et al., 2017), GradCAM (Selvaraju et al., 2016), XRAI (Kapishnikov et al., 2019), and Smoothgrad (Smilkov et al., 2017). Along with using Spearman rank correlation to compare maps before and after model weight and label randomization, we leverage the ground-truth bounding box coordinates

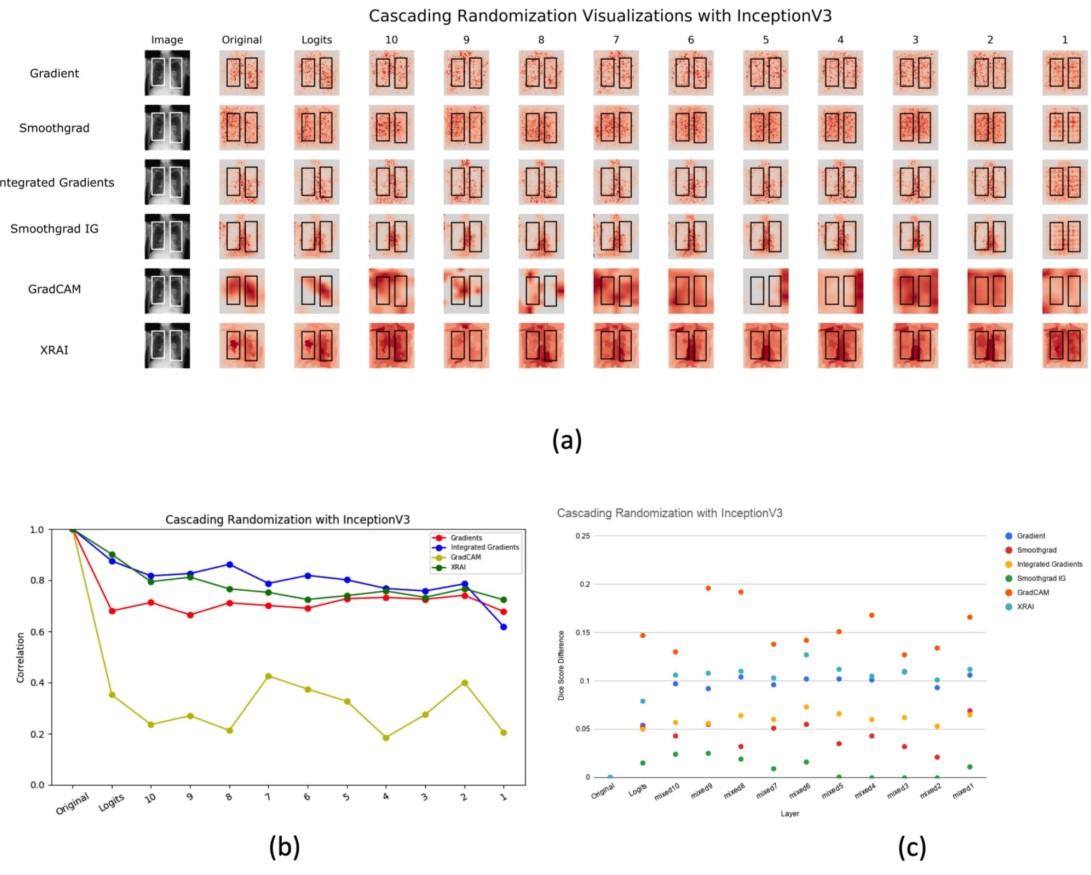

Figure 1: a) Visualization of saliency maps under cascading randomization on InceptionV3 (performance before randomization: AUC=0.98, precision=0.92) (b) Dice scores under cascading randomization (c) Spearman rank correlation under cascading randomization

provided in the RSNA Pneumonia dataset to establish a quantitative baseline using the dice metric. To investigate the sensitivity of saliency methods under changes to model parameters, we employ cascading randomization. (Adebayo et al., 2018) We observed that among these saliency techniques, GradCAM degraded with model randomization to a large degree whereas the other methods did not (Fig 1). This is also verified in a label randomization experiment shown in Fig 2(c) wherein we randomly flipped the labels and retrained the model to observe the difference in the dice scores of the saliency maps. In both the tests, it can be observed that gradient explanation, Smoothgrad IG, and XRAI do not degrade significantly under randomization, suggesting an undesirable invariance to model parameters and labels.

## 2.2. Repeatability and Reproducibility

We also conducted repeatability tests on these saliency methods by comparing maps from a) models with the same architecture trained independently (intra-architecture repeatability)

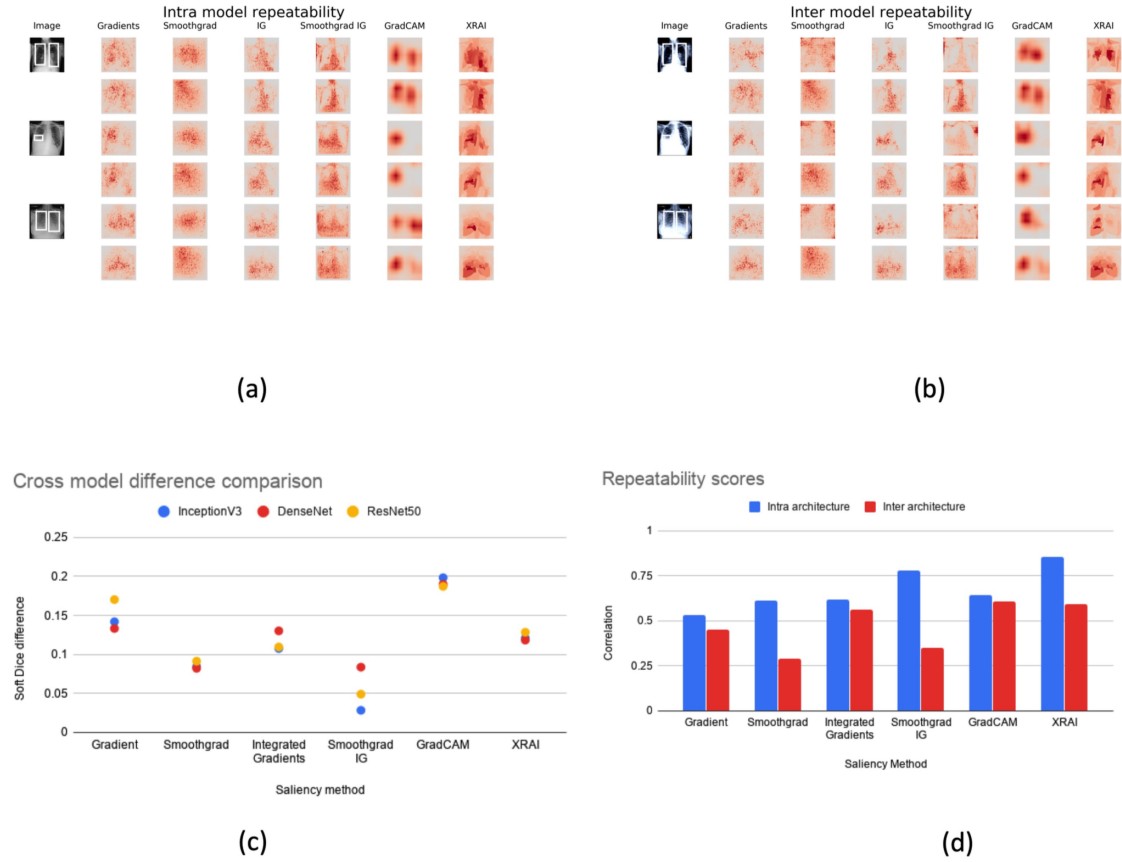

Figure 2: (a) Visualizations from two independently trained InceptionV3 models (b) Visualizations from an InceptionV3 model (top row) and a DenseNet121 model (bottom row) (c) Comparison of dice score differences across saliency methods and architectures (d) Comparison of intra- vs. inter-architecture repeatability using Spearman rank correlation

and b) models with different architectures (inter-architecture reproducibility). These experiments are designed to test if these saliency methods produce similar maps with a different set of weights and whether they are architecture agnostic. Fig 2(a) shows considerable differences across all the different maps from two independently trained InceptionV3 models. Furthermore, Fig 2(b) shows saliency maps differences between those produced from InceptionV3 (top row) versus those from DenseNet121 (bottom row). Fig 2(d) demonstrates that both the Smoothgrad and Smoothgrad IG yielded the most dissimilar maps across architectures while GradCAM yielded maps that were most similar.

## 3. Discussion and Conclusion

In this study, we evaluated the performance of several popular saliency methods on the RSNA Pneumonia Detection dataset in regards to their localization capabilities, robustness to model parameter and label randomization, as well as repeatability and reproducibility with model architectures. It was found that GradCAM showed superior sensitivity to model parameter and label randomization, and was highly agnostic to model architecture. In future studies, we will further examine the effect of different model architectures on saliency maps and validate our findings on a separate medical imaging dataset.

## Acknowledgments

We would like to thank Julius Adebayo for providing us with the cascading randomization code used in his work. (Adebayo et al., 2018)

Research reported in this publication was supported by a training grant from the National Institute of Biomedical Imaging and Bioengineering (NIBIB) of the National Institutes of Health under award number 5T32EB1680 to K. Chang and J. B. Patel and by the National Cancer Institute (NCI) of the National Institutes of Health under Award Number F30CA239407 to K. Chang. The content is solely the responsibility of the authors and does not necessarily represent the official views of the National Institutes of Health.

This publication was supported from the Martinos Scholars fund to K. Hoebel. Its contents are solely the responsibility of the authors and do not necessarily represent the official views of the Martinos Scholars fund.

This study was supported by National Institutes of Health (NIH) grants U01 CA154601, U24 CA180927, and U24 CA180918 and National Science Foundation (NSF) grant NSF 1622542 to J. Kalpathy-Cramer.

This research was carried out in whole or in part at the Athinoula A. Martinos Center for Biomedical Imaging at the Massachusetts General Hospital, using resources provided by the Center for Functional Neuroimaging Technologies, P41EB015896, a P41 Biotechnology Resource Grant supported by the National Institute of Biomedical Imaging and Bioengineering (NIBIB), National Institutes of Health.

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
