# OpenReview forum: "Assessing the validity of saliency maps for abnormality localization in medical imaging"
_MIDL.io/2020/Conference — MIDL 2020_

### Official Review · AnonReviewer1 · 2020-03-13
**Assessing saliency map validity**

**Rating:** 3
**Confidence:** 2

**Review:**

This paper concerns the assessment of saliency map validity.  It was shown that GradCAM is superior to other methods in terms of model and parameter randomization. This is a useful results, as the interpretability that saliency mapping enables is becoming more and more important to help visualize why deep networks are making their decisions. However, there was a lack of discussion of these results in this paper - are there any possible explanations for why GradCAM is performing better? Furthermore, the images in the figures hard to see. They should be larger and as much whitespace should be removed between images.

---

### Official Review · AnonReviewer4 · 2020-03-13
**A study most needed**

**Rating:** 4
**Confidence:** 4

**Review:**

CNN interpretability methods are used more and more in medical image analysis. The authors present a thourough evaluation of several of these methods (localisation capabilities, robustness to model parameter and label randomisation, repeatability and reproducibility with model architectures) extending the work first proposed by Adebayo et al.. This work is very interesting and was most needed.

---

### Official Review · AnonReviewer2 · 2020-03-19
**Fair Comparison of Methods for Model Interpretability for 2D Chest X-Ray Classification**

**Rating:** 3
**Confidence:** 5

**Review:**

The paper compares different state-of-the-art approaches for visual interpretability in 2D chest X-ray classification. The comparison was made based on their localization capabilities, robustness to model parameter, label randomization, and repeatability/reproducibility with model architectures. And the abnormality localization is evaluated with the Dice's score.
1. The paper is well-written and well-organized.
2. The submission relates to the application of deep learning in the field of chest X-ray classification, which is highly relevant to the MIDL audience.
3. The proposed method is technically sound.
4. Experimental results support the claim made in the paper.

---

### Meta-Review · Area_Chair1 · 2020-04-07
**MetaReview of Paper107 by AreaChair1**

**Rating:** 3

**Metareview:**

The interpretability of deep learning models is an important area of research. This work evaluates the usefulness of several methods that aim to visualize the decision making of a neural network. The reviewers are in agreement that the results presented here are of enough interest to warrant acceptance.

**Paper Type:**

validation/application paper

---

### Decision · Program_Chairs · 2020-04-11

Accept